# Discovery of VIP236, an αvβ3-Targeted Small-Molecule–Drug Conjugate with Neutrophil Elastase-Mediated Activation of 7-Ethyl Camptothecin Payload for Treatment of Solid Tumors

**DOI:** 10.3390/cancers15174381

**Published:** 2023-09-01

**Authors:** Hans-Georg Lerchen, Beatrix Stelte-Ludwig, Melanie Heroult, Dmitry Zubov, Kersten Matthias Gericke, Harvey Wong, Melanie M. Frigault, Amy J. Johnson, Raquel Izumi, Ahmed Hamdy

**Affiliations:** 1Vincerx Pharma GmbH, 40789 Monheim, Germany; beatrix.stelte-ludwig@vincerx.com; 2Bayer AG, 65926 Frankfurt, Germany; melanie.heroult@bayer.com; 3Bayer AG, 42096 Wuppertal, Germany; dmitry.zubov@bayer.com (D.Z.); kerstenmatthias.gericke@bayer.com (K.M.G.); 4Vincerx Pharma, Inc., Palo Alto, CA 94306, USA; harvey.wong@vincerx.com (H.W.); melanie.frigault@vincerx.com (M.M.F.); amy.johnson@vincerx.com (A.J.J.); raquel.izumi@vincerx.com (R.I.); ahmed.hamdy@vincerx.com (A.H.)

## Abstract

**Simple Summary:**

The goal of this work was to improve the tumor selectivity of chemotherapy to reduce side effects and improve efficacy. We designed a drug called VIP236 that delivers chemotherapy directly to tumors due to a specialized “homing feature” that binds to αvβ3 integrins. Solid tumors express high levels of αvβ3 integrins. In contrast, expression of αvβ3 is low in healthy tissues. This difference in expression levels leads to preferential homing of the chemotherapy to tumor cells over healthy tissue to reduce side effects. In human cancer models conducted in mice, VIP236 was shown to substantially accumulate in tumors compared with normal tissue. The high accumulation of VIP236 in the tumors resulted in high and long-lasting tumor regression and reduced metastasis formation in brain and lung in cancer models.

**Abstract:**

The emerging field of small-molecule–drug conjugates (SMDCs) using small-molecule biomarker-targeted compounds for tumor homing may provide new perspectives for targeted delivery. Here, for the first time, we disclose the structure and the synthesis of VIP236, an SMDC designed for the treatment of metastatic solid tumors by targeting αvβ3 integrins and extracellular cleavage of the 7-ethyl camptothecin payload by neutrophil elastase in the tumor microenvironment. Imaging studies in the Lewis lung mouse model using an elastase cleavable quenched substrate showed pronounced elastase activity in the tumor. Pharmacokinetics studies of VIP236 in tumor-bearing mice demonstrated high stability of the SMDC in plasma and high tumor accumulation of the cleaved payload. Studies in bile-duct-cannulated rats showed that biliary excretion of the unmodified conjugate is the primary route of elimination. Treatment- and time-dependent phosphorylation of H2AX, a marker of DNA damage downstream of topoisomerase 1 inhibition, verified the on-target activity of the payload cleaved from VIP236 in vivo. Treatment with VIP236 resulted in long-lasting tumor regression in subcutaneous patient-derived xenograft (PDX) models from patients with non-small-cell lung, colon, and renal cancer as well as in two orthotopic metastatic triple-negative breast cancer PDX models. In these models, a significant reduction of brain and lung metastases also was observed.

## 1. Introduction

Following Paul Ehrlich’s vision of a magic bullet (“Zauberkugel”), intense research efforts are ongoing to improve the therapeutic index of antineoplastic agents or radionuclides by restricting their systemic delivery to tumor tissue [1]. With the approval of 13 antibody drug conjugates (ADCs), these are now among the fastest growing drug class in oncology [2,3,4,5]. The ADC mode of action is well understood and follows the route of target binding mediated by the antibody and subsequent internalization and trafficking of the ADC to the lysosomes where the active metabolite is released. The expansion of payload classes beyond tubulin binders by employing topoisomerase I (TOP1) inhibitors as payloads in ADCs such as trastuzumab deruxtecan (Enhertu^®^) [6] and sacituzumab govitecan (Trodelvy^®^) [7] was one of the drivers of recent successes. However, clinical use of ADCs is limited by substantial toxicity in patients, and dose-limiting toxicities are often shared across the same cytotoxic payload regardless of the antibody target. Clinical evidence suggests that the tolerated doses of ADCs are not significantly different from those of related small molecules [8].

The emerging field of small-molecule–drug conjugates (SMDCs) for targeted delivery is less established but has the potential to overcome some of the ADC-related issues. Due to a significantly reduced molecular size, SMDCs are non-immunogenic and may have an improved penetration into tumor tissue. Furthermore, synthetic accessibility is less complex and less expensive [9,10]. Examples of targets explored for SMDC delivery with small molecule or peptide ligands include the folate receptor [11], αvβ3 integrin [12,13,14,15,16], carbonic anhydrase IX [17], somatostatin receptor 2 [18], fibroblast activation protein [19], ephrin A2 [20], sortilin [21], and nectin 4 [22]. Looking for targets that are broadly expressed in solid tumors and associated with aggressive disease, we were particularly interested in αvβ3 integrins. αvβ3 Integrins are heterodimeric transmembrane receptors that are overexpressed on activated endothelial cells [23] and on several tumor cell types including metastatic tumors [24]. These receptors bind to arginine–glycine–aspartate (RGD)-containing matrix proteins and have an important function in tumor-induced angiogenesis. In the tumor microenvironment (TME), αvβ3 integrins play critical roles in tumor progression, resistance to cytotoxic therapy, metastasis, and the recruitment of immune and inflammatory cells [25]. Expression of αvβ3 integrin is associated with poor prognosis in melanoma [26], cell invasion in hepatocellular carcinoma [27], and metastasis of chondrosarcoma [28].

Neutrophil elastase (NE) belongs to the serine family of proteases and degrades elastin and other extracellular matrix proteins, which contribute to cancer progression by enhancing tumor evasion and metastasis [29]. It is expressed by neutrophils, macrophages, and tumor stroma and is found elevated in lung, breast, and kidney tumors but remains low in normal tissues [30]. Elastase expression and neutrophil tumor infiltration have been correlated with metastatic potential and poor prognosis. Survival of patients with breast cancer and non-small-cell lung cancer (NSCLC) is poor in those individuals expressing high NE as compared with those expressing low NE levels [31]. ADCs with NE-sensitive linkers are currently being explored [32,33].

We recently described a novel SMDC designed for αvβ3-targeted delivery of an optimized camptothecin payload to cancer cells with NE-mediated payload release, as illustrated in Figure 1A [34]. Efficient tumor homing of the peptidomimetic integrin binder employed in the SMDC was demonstrated in imaging studies. Furthermore, we showed in vitro data of the optimized camptothecin payload supporting the potential to overcome transporter substrate liabilities associated with resistance to SN38, the active metabolite of irinotecan. The favorable PK profile resulted in a robust anticancer activity in cell-line-derived models representing different tumor indications [34].

Herein, for the first time, we disclose the synthesis and structure of the SMDC VIP236 consisting of the three modules outlined in Figure 1B and further elucidate VIP236 pharmacokinetics, pharmacodynamics, and activity in patient-derived xenograft models including metastatic models.

## 2. Material and Methods

### 2.1. Imaging Study Showing Elastase-Activity in Tumors

To confirm the presence of free active NE in tumors, an elastase substrate peptide labelled by infrared fluorophore and corresponding quencher (Lys(QSY21)-ANPV-Cys(AlexaFluor647)) was ordered for custom synthesis at Anaspec, Fremont, CA, USA. Cleavage of the substrate by NE leads to a strong increase in AlexaFluor647 fluorescence at 680 nm after excitation at 625 nm. Then, 200 nmol of the quenched peptide dissolved in 100 µL phosphate-buffered saline (PBS) was injected into the tail vein of anesthetized mice carrying a Lewis lung tumor of around 1500 mm^3^ volume. Mice were imaged in the infrared imager (Kodak In Vivo Multispectral Imaging System FX, Kodak, Rochester, NY, USA) 10 and 90 min after injection.

### 2.2. Compounds

The SMDC VIP236 (**8**) and its epimers **8e1** and **8e2** were synthesized in the Medicinal Chemistry Research Labs of Bayer AG, Pharmaceuticals, Wuppertal, as outlined in the Results Section 3.2, with further details and analytical data provided in the Appendix A.

### 2.3. Stability in Rat Plasma and in Buffer at pH 7.4

The SMDC VIP236 was dissolved in acetonitrile/DMSO (1:1, 0.5 mL). While vortexing, 20 µL of this solution was added to 1 mL 37 °C rat plasma. The incubation was stopped at respective time intervals by adding acetonitrile/buffer pH3 (80:20) and VIP236, and formation of degradation products was analyzed by LC and LC/MS.

For hydrolytic stability measurement, VIP236 was dissolved in PBS buffer at pH 7.4 in a concentration of 5 mg/mL, and the solution was stored at 4 °C. LC/MS were measured after 2, 6, 30, and 79 days.

### 2.4. Metabolic Stability in Hepatocytes

VIP236 (1 µM) was incubated separately in duplicate in a 24-well plate containing mouse, rat, dog, or human hepatocyte suspensions (1 million cells/mL) at 37 °C for 4 h under 5% CO_2_:95% air. Aliquots (50 μL) were taken at 0, 15, 30, 60, 120, 180, and 240 min, and the samples were treated with acetonitrile (200 μL) containing an internal standard (100 ng/mL tolbutamide). The samples were centrifuged (4000 rpm for 10 min), and aliquots (100 μL) of the supernatant were taken and mixed with water (100 μL). A 5 μL aliquot was analyzed by LC/MS/MS for loss of VIP236.

### 2.5. Human and Mouse NE Biochemical Cleavage Assay

Different volumes of VIP236 were added to microtiter plate wells to final concentrations ranging from 0–50 μM. Then, 125 μL of pre-heated buffer (100 mM Tris pH 7.5, 150 mM NaCl, 10 mM CaCl_2_, and 0.05% BSA) was added to each incubation. The reaction was started by adding 0.5 or 1 nM activated mouse elastase (according to manufacturer’s protocol) or human NE to the reaction vials. A negative control without elastase was run under the same conditions. The reaction was stopped at 0, 4.5, 8, and 12 min by the addition of acidified acetonitrile/0.2% formic acid. Camptothecin was used as internal standard for the LC/MS/MS analysis. The kinetic parameters describing the cleavage of VIP236 by NE were estimated by fitting to the Michaelis–Menten equation.

### 2.6. In Vitro Proliferation Assay in the Presence and Absence of NE

The in vitro cytotoxicity was evaluated in cancer cell lines NCI-H292 and LoVo after a 72-h continuous exposure to the presence or absence of 20 nM NE using MTT assays (ATCC, Manassas, VA, USA). IC_50_ values were determined as the concentration of compound required for 50% inhibition of cell viability.

### 2.7. In Vivo Pharmacokinetics in Tumor-Bearing Mice

Female NMRI nu/nu mice bearing 786-O tumors were given a 4 mg/kg VIP236, **8e1** (VIP407), or **8e2** (VIP940) dose or an equimolar 1 mg/kg dose of **1** intravenously (IV) via tail vein. Following delivery of VIP236, the control epimers **8e1** or **8e2**, or unconjugated payload **1**, tumor-bearing mice (n = 2 per timepoint) were sacrificed at pre-dose, 0.033, 0.167, 0.5, 1, 2, 4, 7, and 24 h post dose. Plasma and tumor samples were collected and frozen at <−15 °C until sample analysis. Samples were analyzed for concentrations of respective SMDCs and the released payload **1** by LC/MS/MS. Pharmacokinetics of VIP236 [34] in comparison to **8e1**, **8e2**, and payload **1** [34] were determined using geometric mean concentration–time profiles. All pharmacokinetic parameters were calculated by non-compartmental methods [35].

### 2.8. VIP236 Bile-Duct-Cannulated Rat Study 

Male bile-duct-cannulated rats (n = 3) were administered a 10 mg/kg intravenous bolus dose of VIP236 via a jugular vein catheter. Bile was collected from 0–4, 4–8, 8–24, and 24–48 h after dosing; urine was collected from 0–8, 8–24, and 24–48 h after dosing; and feces were collected from 0–24 and 24–48 h after dosing. LC/MS/MS was used to quantify VIP236 and payload **1** in samples of bile and urine. Feces were not analyzed, as there was complete recovery of the entire VIP236 dose in bile and urine.

### 2.9. IHC Analysis

At least 50 patient-derived samples for each reported indication were evaluated for specific αvβ3 staining using anti-αvβ3 antibody (monoclonal/Rabbit/Clone EM22703, Sigma; 1:50 dilution, incubation time 32 min at room temperature [RT]). For staining, the Detection kit Ultra View Universal DAB (Ventana, Oro Valley, AZ, USA) was used (incubation time 8/8/4 min incubation at 37 °C). For evaluating the presence of human NE, the tissue samples were evaluated using anti-NE antibody (Abcam (Boston, MA, USA)/ab68672; 1:150 dilution, incubation time 20 min at RT). For staining, the Bond Polymer Refine Detection kit (Leica (Wetzlar, Germany)/Menarini (Florence, Italy)) was used (incubation time 8/8/10 min at RT). All samples were evaluated by a pathologist.

The tissues of 5 animals per group were examined according to standard protocol and in parallel evaluated for necrotic areas and connective tissue. The analysis was restricted to viable neoplastic cells. The semiquantitative IHC-scoring scale (0 to 3) was related to percent staining of neoplastic cell (0: no stain; 1: up to 25% of the cells show positive immunoreactivity; 2: 25–50% of the cells show positive immunoreactivity; 3: 50–100% of cells show positive immunoreactivity) to deduce a composite score. Light microscopic images acquired at 200× magnification and evaluation of 10 high-power fields were averaged for each sample. Antibodies used: TOP1 antibody (Abcam Ab109374, dilution 1:100 in PBS), incubation 60 min at room temperature (RT); αvβ3 antibody (Bioss bs-1310R, Woburn, MA, USA, dilution 1:200 in PBS) incubation time 60 min at RT; NE antibody (Booster Bio PB10058, Pleasanton, CA, USA, dilution 1:100 in PBS: incubation time overnight at 4 °C). A positive and negative tissue control for the respective marker was included in the staining process. Counterstain was performed with Mayer’s hematoxylin with a 1 min staining time at RT. 

### 2.10. H2AX Staining as a Marker for DNA Damage in SNU16 CDX Mouse Model

A total of 72 xenograft tumor samples were analyzed for their content of γH2AX by means of a validated IHC assay (Nuvisan, Neu-Ulm, Germany). All samples were fixed in 10% neutral buffered formalin immediately after tumor excision and subsequently stored in 70% ethanol at 4 °C and processed into paraffin after 24 h of fixation. Mouse anti-γH2AX (Merck Ab#05-636, Darmstadt, Germany) was used with Envision System and DAB+ Chromogen/Substrate Buffer (Dako/Agilent, Santa Clara, CA, USA) for analysis of γH2AX IHC. A ScanII slide scanner was used to digitize IHC images for quantification (QuPath; https://qupath.github.io (accessed on 20 August 2023)) of the percentage of γH2AX-positive cells and represented in a histogram with standard deviation.

### 2.11. PDX Models

Studies were conducted according to all applicable international, national, and local laws and followed the national guidelines for the Care and Use of Laboratory Animals of the Society of Laboratory Animal Science (GV-SOLAS). All animal experiment protocols were approved by the Regional Council Committee on the Ethics of Animal Experiments. Non-small-cell lung (LXFL529) and renal (RXF2667) PDX mouse models comprised six groups each with 10 animals. Tumor fragments (3–4 mm edge length) were transplanted, and tumor growth was monitored until a median size of about 120–150 mm^3^ was achieved. VIP236 was administered intravenously in a 3 days on/4 days off (vehicle, 20 and 40 mg/kg) or 2 days on/5 days off (30 and 40 mg/kg) dosing schedule as well as once weekly (60 mg/kg). Both experiments were terminated after a three-week recovery phase on day 63. Animals were monitored at least twice daily on working days and at least once daily on weekends (tumor size and body weight measurement). Colon (CXF2068) and triple-negative breast cancer (TNBC) (MAXF BR120) PDX mouse models were performed as described above with changes in schedule. VIP236 was administered in a 2 days on/5 days off (vehicle, 20 mg/kg and 40 mg/kg), once weekly (60 mg/kg), or 2 days on/12 days off (40 mg/kg) schedule. The statistical significance of antitumor efficacy was analyzed by the non-parametric Kruskal–Wallis test, followed by Dunn’s method for multiple comparisons using GraphPad Prism bioanalytic software (version 9.10), with *p* < 0.05 as significant. Additionally, a nonpaired *t*-test was performed using GraphPad Prism (version 9.10).

### 2.12. Orthotopic TNBC Model MA4296 and MA15191

Two orthotopic TNBC PDX mouse models were performed. Tumor cells were inoculated into the mammary fat pad of mice. Therefore, the mice were anesthetized shortly before surgery. After disinfection of the skin, mice received a small incision in the skin between the 4th and 5th dug. Each mouse was implanted with 2.5 × 10^6^ (MA4296) or 5 × 10^6^ (MA15191) cells. Treatment started on day 20 after inoculation, when palpable tumor sizes reached 127 mm^3^ (day 33 for MA15191). VIP236 was administered via intravenous injection with 40 mg/kg of VIP236 at two successive days followed by 5 days off or with 60 mg/kg of VIP236 once weekly for 4 weeks. Tumors were measured twice weekly with calipers, and their volume was calculated with the following formula: (length × width^2^)/2. The body weight was also determined two times weekly. Reasons for termination were tumor size > 1.5 cm^3^ or poor general condition. At the end of the study, tissue sampling for IHC and PCR analyses was performed. The statistical analysis was performed using mixed-effect analysis with Tukey’s multiple comparisons test (GraphPad Prism 8.4.2) to calculate significance in tumor growth inhibition between vehicle and treatment groups over time. The Kruskal–Wallis test was used to determine significance at study days 43 and 47. Additionally, the unpaired *t*-test was performed.

### 2.13. PCR Analysis

To detect human tumor cells, a human-specific RealTime DNA PCR using an optimized primer-probe-assay targeting the alpha-satellite region of the human chromosome 17—a locus, which is sufficiently different from similar loci in the mouse genome—was performed [36]. Genomic DNA was isolated from liver, lung, and brain of all mice per group using the Qiagen Dneasy Blood & Tissue Kit (Qiagen GmbH, Hilden, Germany) as recommended by the manufacturer.

For RealTime PCR, the ready-to-use primer-probe-assay and the TaqMan^®^ Universal PCR Master Mix, No AmpErase^®^ UNG (Applied Biosystems GmbH, Weiterstadt, Germany, Prod-No. 4324018) were used. The PCR reaction and quantification were performed in a StepOnePlus™ RealTime PCR System (LightCycler 480 II system (Hoffmann-La Roche, Basel, Switzerland) applying the standard protocol supplied by the manufacturer. For semi-quantitative evaluation, the crossing point (Cp) value, also known as cycle threshold, was used. Cp values correlate inversely with the DNA load, with values ≥ 35 considered negative. Genomic DNA from an appropriate human tumor xenograft as a positive control, from NOD/SCID mouse liver as a negative control, as well as a water sample as a reagent control were processed in parallel. The statistical analysis was performed using GraphPad Prism version 8.4.2 for Windows, GraphPad Software, La Jolla, CA, USA.

## 3. Results

### 3.1. Tumor-Associated NE Cleavage

The presence of free and active NE in tumors was evaluated in the Lewis lung carcinoma tumor model by IV injection of the fluorescently labelled elastase-specific substrate. The elastase recognition sequence (NPV) in this quenched substrate was very similar to elastase sensitive peptide linker DPV, which was employed in VIP236 with comparable elastase cleavage efficacy and selectivity. Substrate cleavage corresponding to the increase in the fluorescent signal was monitored in the infrared fluorescent imager at excitation wavelength of 625 nm and emission wavelength of 680 nm. Strong fluorescent signal was observed in tumor tissue 10 and 90 min after injection, and lower/no detectible substrate cleavage in other tissues and organs indicated increased NE enzymatic activity in the tumor tissue. Superposition of X-ray and fluorescent images are depicted in Figure 2.

### 3.2. Synthesis of VIP236 and Epimers

The small-molecule–drug conjugate VIP236 was designed to target αvβ3 integrins in tumor tissue and to release the payload 7-ethyl camptothecin **1** [37,38] upon enzymatic cleavage by NE in a traceless manner. The RGD-peptidomimetic αvβ3 ligand **6** has shown efficient tumor homing when coupled to a fluorescent dye [34]. In the SMDC VIP236, this integrin ligand **6** is now attached via a short PEG-spacer to a tripeptide L-aspartic acid-L-proline-L-valine, which at the C-terminal end is linked through an ester bond to the hydroxy-lactone ring of the 7-ethyl-camptothecin payload **1 [37,38]**. The 7-ethyl camptothecin payload **1** was selected to meet the physicochemical and permeability properties tailored for extracellular release in the TME and to address liabilities of SN38, the active metabolite of irinotecan and sacituzumab govitecan [7,39]. The linker peptide was selected to be a substrate sequence of NE and at the same time to provide maximal stability of the ester bond connecting the payload due to steric hindrance. The synthesis of VIP236, consisting of three modules, namely the peptidomimetic αvβ3 binder **6**, the NE cleavable linker, and the 7-ethyl camptothecin payload **1**, is outlined in Figure 1. As additional tool compounds suitable to investigate the contribution of the individual modules of VIP236 to the payload delivery to tumor cells, the two most relevant epimers **8e1** and **8e2** were synthesized following the same route (Figure 1). 

For an efficient NE-mediated release of the active payload **1** from the SMDC VIP236, the ester bond between L-valine and payload **1** needs to be cleaved at the C-terminal end of the linker peptide aspartic acid-proline-valine, which was found to be a substrate tripeptide of NE. Replacement of L-valine in the SMDC VIP236 by the unnatural amino acid D-valine provided the epimer **8e1**, deemed to be a non-cleavable SMDC, which served as a control to show the impact of elastase cleavage for activity. 

In the control SMDC **8e2**, the peptidomimetic integrin ligand **6e2** with S-configuration at the stereocenter instead of R-configuration in **6** was employed, which has a major impact on the affinity to αvβ3 integrins. **8e2** represents a weakly binding epimer with ~25-fold reduced binding affinity to αvβ3 integrins as compared with VIP236 [40]. Therefore, **8e2** is an appropriate control to investigate the contribution of the integrin-binding moiety **6** to the tumor homing of the SMDC VIP236.

The attachment of a valine residue to the 20-hydroxy group of 7-ethyl camptothecin is challenging due to steric hindrance and the risk of epimerization; it was accomplished as outlined in Figure 1 by coupling of the activated building block Boc-valine-N-carboxy anhydride **2** in dichloromethane in the presence of DMAP (a). Subsequent deprotection (b) and acylation with the partially protected linker peptide **4** in the presence of EDCI, Oxyma, and diisopropylethylamine in DMF (c), followed by removal of the protecting groups with TFA in dichloromethane (d), provided the intermediate **5**. Coupling of **5** with the activated integrin ligand **7** in the presence of diisopropylethylamine in DMF (f) and subsequent transformation in acetone/water to the bis-sodium salt (g) provided the SMDC VIP236. Details of the synthesis outlined in Figure 1 and analytical characterization are provided in the Appendix A. The epimers **8e1** and **8e2** were synthesized in analogy, starting with the enantiomers **2e1** instead of **2** and **6e2** instead of **6**, respectively. Enantiomeric purity of **7** and **7e2** is given in Appendix A.

### 3.3. In Vitro Evaluation: VIP236 Is Highly Stable but Efficiently Cleaved by Mouse and Human NE

For efficient delivery of 7-ethyl camptothecin payload **1** to tumor tissue, high stability of the SMDC in circulation and as well as efficient cleavage by NE is critical. VIP236 is highly stable when dissolved in PBS buffer at a concentration of 5 mg/mL, with no degradation observed after 79 days of storage in solution at 4 °C (LC/MS measured after 79 days is shown in Appendix A). Furthermore, upon incubation in rat plasma at 37 °C, VIP236 was stable for over 24 h (Appendix A). VIP236 was also found to be very stable (>80% of parent drug remaining) in mouse, rat, dog, and human hepatocytes after a 4-h incubation. 

The kinetic parameters describing the cleavage of SMDC VIP236 to payload **1,** i.e., the Michaelis–Menten constant (Km) and maximum turnover number (kcat) by human and murine NE, were determined at enzyme concentrations of 0.5 and 1.0 nmol. The kinetic parameters estimated for the two enzyme concentrations were determined for the following:

Human NE (Appendix A):Km = 9.4 and 9.6 μM, respectively;kcat = 593 and 691 1/min, respectively.Mouse elastase (Appendix A):Km = 15.9 and 8.9 μM, respectively;kcat = 94 and 80 1/min, respectively.

Overall, Km of NE was similar when comparing the mouse and human enzyme.

To evaluate the requirement for NE to cleave the active payload, we evaluated the in vitro cytotoxic activity of VIP236 and respective epimers **8e1** and **8e2** in comparison to the payload **1** itself in NCI-H292 and LoVo cancer cell lines in the presence or absence of NE. Without NE, all SMDCs showed weak cytotoxic activity with IC_50_ values in the three-digit nanomolar range. Only when NE was added to the culture medium the cytotoxic activity of VIP236 and its weakly binding epimer **8e2** increased by about 25–100-fold to reach similar potency as the payload **1** alone in contrast to the non-cleavable epimer **8e1** (Table 1 and Appendix A).

### 3.4. In Vivo Pharmacokinetics and Bile Duct Cannulated Rat Study

VIP236, **8e1**, **8e2**, and payload **1** plasma and tumor concentration–time profiles after intravenous administration are presented in Figure 3. Pharmacokinetics parameters for the SMDCs are shown in Appendix A. Clearance (CL) was low for all three SMDCs in tumor-bearing mice, with VIP236 CL < **8e2** CL < **8e1** CL. The volume of distribution at steady state (V_ss_) was low for all three SMDCs and approximated plasma volume for VIP236 and **8e2** [41]. For **8e1**, V_ss_ was slightly larger, as it was approximately two times the plasma volume. Half-life was short for all compounds (~1 h).

After a single IV administration of VIP236, **8e2**, and **8e1** at 4 mg/kg, concentrations of the parent SMDCs were higher in plasma compared with tumor (Figure 3A,C,E). Delivery of payload **1** to the tumor was dependent on which SMDC was administered. When delivered via VIP236, the AUC_tumor_/AUC_plasma_ ratio of released payload **1** was 6.1, which is ~11-fold higher compared to the AUC_tumor_/AUC_plasma_ ratio of 0.61 measured after direct IV administration of payload **1** [34] and suggests that delivery of payload **1** via VIP236 concentrates the payload **1** in tumor tissue (Table 2). The weakly binding epimer **8e2** did not appear to preferentially deliver payload **1** to tumor tissue, while payload **1** was not detected in tumors after administration of the non-cleavable epimer **8e1**.

The recovery of the VIP236 dose in male bile-duct-cannulated rats is shown in Table 3. In non-tumor-bearing rats, almost the entire dose was recovered as unchanged VIP236 in bile (~100%) (Table 3). Small quantities of the VIP236 dose were recovered as unchanged VIP236 in urine (~2.4%). Only trace levels of cleaved payload **1** were quantified in bile and urine (<1% total;). Overall, the main route of elimination of VIP236 in rats appears to be via biliary excretion of unchanged VIP236.

### 3.5. IHC Staining Confirms αvβ3 and NE Presence in Patient Tumor Samples

Patient tumor samples (≥50) from different cancer indications were evaluated to confirm αvβ3 and NE presence in advanced cancers where clinical data indicate correlation of NE expression and poor prognosis [31]. Poor survival also correlates with αvβ3 expression [42,43]. NE staining as well as αvβ3 staining on endothelial cells was detected, as shown in Figure 4. In some indications, additional membranous staining of αvβ3 on tumor cells was observed (e.g., renal cancer).

### 3.6. Time- and Dose-Dependent Induction of DNA Damage by VIP236

TOP1 inhibitors deliver DNA damage, which can be measured by phosphorylated H2AX (γH2AX) [44]. Therefore, the pharmacodynamic effect of the payload **1** of VIP236 was evaluated in an SNU16 (gastric cancer cell line) xenograft mouse model. As a certain amount of baseline DNA damage is expected, the pretreatment samples had 36% γH2AX-positive cells. A time-dependent increase in the percentage of γH2AX-positive cells was detected by IHC after treatment on day 2 (Figure 5A). At the 20 mg/kg dose level, robust induction of γH2AX was observed with a range of 68% to 79% γH2AX-positive cells between the 24.25–48 h timepoints (Figure 5B). The increase in γH2AX returned to baseline levels 144 h after treatment. For some of the groups (e.g., the 144 h time point of the 20 mg/kg dose group), not enough tumor volume was available for preparation of samples for γH2AX IHC analysis since tumor volume had greatly decreased after one week of treatment.

### 3.7. Monotherapy of VIP236 Achieved Potent and Durable Antitumor Activity in PDX Tumor Mouse Models

VIP236 as monotherapy was investigated in the taxane-resistant PDX lung tumor model LXFL 529 (NSCLC), the clear-cell renal carcinoma tumor model RXF 2667, the liver metastasis of colon tumor model CXF2068, and the TNBC model MAXF BR120, all implanted subcutaneously in NMRI nude mice. 

The LXFL 529 model was derived from the primary tumor of a 34-year-old woman with unknown treatment history. In this model, tumor volumes in all therapeutic groups were significantly reduced compared with the control group on day 35 (*p*-values < 0.0001) (Figure 6A). Tumor volumes in the re-growth phase demonstrated a long-lasting effect of antitumor efficacy. In this model, 40 mg/kg 3 days on and 4 days off was the most efficacious regimen based on an 100% overall response rate (80% complete response + 20% partial response) (Figure 6E). 

The RXF 2667 model was derived from a clear-cell carcinoma of a 61-year-old woman with unknown treatment history. Monotherapy with VIP236 achieved high antitumor efficacy in all treatment groups, including sustained antitumor effect during the regrowth period. (Appendix A). The CXF 2068 model was derived from liver metastasis from a primary colon cancer tumor of a 50-year-old, previously untreated man. In this model, VIP236 monotherapy at 20/40 mg/kg (2 on/5 off) or at 60 mg/kg (once weekly) reduced tumor volumes significantly in comparison to the control vehicle (*p* < 0.001 Kruskal–Wallis (Figure 6C)). Treatment of VIP236, when administered less frequently at 40 mg/kg (2 on/12 off), resulted in borderline antitumor efficacy that did not reach statistical significance. The MAXF BR120 was derived from a primary TNBC tumor from a 52-year-old woman after chemotherapy. All VIP236 monotherapy regimens showed statistically significant antitumor efficacy in this model (Appendix A). VIP236 was well tolerated across PDX models, as evidenced by little weight loss (representative data in Figure 6B,D).

### 3.8. Significant Impact of VIP236 on Primary Tumor and Metastases in Orthotopic Metastatic TNBC PDX Mouse Models

MA4296 was established from a 43-year-old woman with relapsed squamous metastatic TNBC. MA15191 was derived from a 53-year-old woman with an invasive triple-negative ductal breast carcinoma. Mutation analysis revealed KRAS wild type for both models. The impact of VIP236 monotherapy on tumor growth is shown in Figure 7A,C. Treatment with 40 mg/kg VIP236 (2 on/5 off) resulted in significant tumor growth inhibition of MA4296 (optimum T/C: 4%, day 46, *p* < 0.01; partial remission) and of MA15191 (T/C: 17%, *p*-value 0.015, Appendix A). When given at 60 mg/kg once per week, VIP236 also significantly inhibited orthotopic growth of MA4296 but with reduced efficacy (T/C: 36%, day 36, *p* < 0.05; stable disease) (Table 4). Treatment with both schedules of VIP236 was well tolerated. No early death events occurred, and all mice were in a good general condition, as indicated by the lack of body weight loss (Figure 7B,D).

These PDX models were also chosen to investigate the anti-metastatic effect of VIP236. After termination of the study, mice were inspected for pathologic signs of treatment and for macroscopic metastases at all organs. To look for micrometastasis in lung, brain, and liver tissues, material was prepared from vehicle and treated animals for PCR analyses.

Human DNA was found in lung tissue (moderate expression), brain tissue (low expression), and liver tissue (marginal expression) in both models. In all three organ tissues, treatment with VIP236 reduced the metastatic load compared to the vehicle control. Results of the MA4296 PDX model are depicted in Figure 8. In lung tissue, both doses of VIP236 reduced the ratio of human DNA significantly. Treatment with 40 mg/kg (2 on/5 off) decreased the relative DNA expression highly significantly, with a *p*-value of <0.0001 and with 60 mg/kg (weekly); thus, a lower but still statistically significant DNA reduction was achieved (*p* = 0.032). In liver tissue samples, a reduction in human DNA load after treatment with 40 mg/kg VIP236 was observed, which did not reach statistical significance. In brain tissue samples, reductions of human DNA were measured, which were statistically significant with 40 mg/kg (2 on/5 off) doses of VIP236 (*p* = 0.0067) but not with 60 mg/kg (weekly). The antimetastatic effect of VIP236 was reproduced in the MA15191 model. These observations may indicate blood–brain barrier penetration of VIP236 and its toxic payload **1**.

### 3.9. Increase in αvβ3 and NE Staining upon VIP236 Treatment, While TOP1 Expression Is Stable

Due to αvβ3 integrin upregulation during metastasis formation, we specifically investigated the effect of VIP236 treatment in the aforementioned orthotopic TNBC metastasis models [45]. After 4 weeks of treatment with VIP236, the tumors were FFPE-processed to evaluate the presence of αvβ3, NE, and the impact on TOP1 expression. All samples were evaluated by IHC analysis in a semi-quantitative way. The results are summarized in Appendix A. The IHC analysis revealed a rise of αvβ3 presence in the neoplastic cells derived from the orthotopic, metastatic breast cancer PDX model by a higher IHC scoring at treatment end (0.5 to 2.2). Increased infiltration of murine NE occurred from 0.2 to 1.5, while TOP1 expression was stable (3 to 3). These results suggest a positive-feedback loop may occur, producing the durable antitumor efficacy observed in in vivo tumor models. 

## 4. Discussion

VIP236 was designed for targeted delivery of payload **1** to tumors expressing αvβ3 integrins mainly on activated endothelial cells in the TME and for extracellular NE cleavage of the linker to release payload **1**. This design of VIP236 addresses the information gleaned from previous compounds targeting αvβ3 integrins to inhibit angiogenesis, which despite compelling preclinical results, showed only limited efficacy in the clinic, notwithstanding a good safety profile [46,47,48]. Imaging studies and tissue analysis suggested that αvβ3 integrin inhibitors such as cilengitide and volociximab were highly effective at reaching their intended targets [46]. Therefore, our goal was to leverage the efficient homing of αvβ3 binders and use it for the targeted delivery of a cytotoxic agent to the TME. Our approach does not depend on the inhibition of integrin signaling for its therapeutic effect.

We were able to confirm the tumor homing of the peptidomimetic integrin binder employed in the SMDC VIP236 [34] as well as the tumor localization of NE activity in imaging studies in tumor-bearing mice, which consequently supports the design rational. Furthermore, the high expression of both markers in the TME and their association with aggressive disease and poor prognosis for patients suggests the potential of VIP236 as a pan-solid tumor agent in advanced and metastatic cancers. This potential for broad therapeutic applicability is supported by the activity of VIP236 in PDX tumor models across multiple indications, as previously shown in cell-line-derived models derived from breast, colon, renal, and lung cancers [34]. Here, we demonstrated the high potency of VIP236 monotherapy in a broad panel of PDX models, as exemplified by a long-lasting, complete response in the taxane-resistant NSCLC model LXFL529. Due to upregulation of αvβ3 during metastasis formation, we also specifically investigated the effect of VIP236 in the orthotopic TNBC PDX models MA4296 and MA15191. In addition to a regression of the primary tumor, a significant reduction of micrometastasis in lung and brain tissue was observed.

Other αvβ3-targeted drug conjugates described in the literature [12,13,14,15,16] are designed for activation and drug release mediated by cathepsin or by legumain either intracellularly upon internalization or extracellularly under suboptimal pH conditions. Furthermore, self-immolative spacer groups are often required to enable payload release but associated with an increased lipophilicity of the conjugates. We considered an extracellular release of the payload **1** in the TME as an appropriate match to the αvβ3-mediated delivery and envisioned NE as an appropriate cleavage enzyme to further enhance the targeting specificity of VIP236. Based on publicly available data on NE cleavable probes [49] and leveraging our previous experience with camptothecin conjugates [50,51], the SMDC VIP236 was optimized for high aqueous solubility, high stability, and efficient cleavability by NE without a need for insertion of self-immolative spacers. Linker length, the carboxy side chain of aspartic acid, and the sterical hindrance of the valine residue with a stabilizing impact on the ester bond contribute to this favorable profile [50]. Irinotecan is a water-soluble prodrug of the TOP1 inhibitor, SN38, which is activated by carboxylesterase to form its active metabolite with higher cleavage efficiency in rodents as compared with humans [39]. In contrast, no decrease in the efficiency of NE-mediated release of the payload is anticipated with VIP236 based on kinetic studies showing similar affinity (Km) of mouse and human NE. In vitro cytotoxicity of VIP236 was found to be NE-dependent. The stability profile was also reflected in pharmacokinetic studies in tumor-bearing mice, with very low levels of released payload detectable in circulation. The SMDC VIP236 has a shorter half-life in comparison to full-sized antibodies and ADCs, but it still shows low clearance and related high exposure and a small volume of distribution. It still delivers payload **1** efficiently to the tumor with a tumor-to-plasma ratio 11-fold higher after administration of VIP236 as compared with administration of an equimolar dose of payload **1** in 786-O tumor-bearing mice. As shown in a study with bile-duct-cannulated rats, almost the entire dose is recovered as unchanged VIP236 in bile. Thus, the main route of elimination of VIP236 in rats appears to be biliary excretion of unchanged VIP236, which is differentiated from the renal excretion of other peptide–drug conjugates [20]. Fourteen ADCs have received marketing approval, most in the last few years, ushering in a new age in “biologic missiles” for targeted cancer therapy [52]. Nonetheless, the application of ADCs for the treatment of solid tumors faces challenges. For example, the large size of ADCs (~ 150 kDa) can hinder deep penetration into the tumor matrix [53]. This inefficient tumor penetration might be the reason why relatively high levels of ADCs are required to achieve efficacy, and consequently, ADCs have not been found to be less toxic than their cytotoxic counterparts [8,54]. Another limitation is that current ADCs require the cell surface expression of a tumor-specific antigen that must also internalize to release the payload. These factors have currently limited the approval of ADCs to a small number of solid tumors that express Her2, Trop2, nectin4, tissue factor, and folate receptor α [55]. Unlike ADCs, VIP236 is a small molecule of ~1.5 kDa and has shown efficient penetration in tumors and affords more than a 10-fold increase in intratumor/plasma payload exposure ratio compared to direct administration of the payload alone. Also, because VIP236 targets αvβ3 and is cleaved extracellularly by NE in TME, it is expected to have broad activity across many tumor types, especially advanced aggressive tumors with the highest unmet medical need. Taken together, these findings support further investigation of VIP236 in clinical trials as a potential treatment option for patients with advanced or invasive solid tumor cancers.

## 5. Conclusions

VIP236 was designed and optimized for targeting αvβ3 integrins highly expressed on activated endothelial cells in the TME and solid tumors and for extracellular cleavage of an optimized 7-ethyl camptothecin payload **1** by NE. VIP236 shows remarkable chemical, plasma, and metabolic stability. It is selectively cleaved by murine and human NE with similar affinity to release the payload. An 11-fold higher tumor-to-plasma ratio of payload **1** when delivered via VIP236 versus direct IV administration of equimolar amounts of **1** was observed in rats. In contrast, non-cleavable and weakly binding epimers **8e1** and **8e2**, respectively, were not able to accumulate payload **1** in the tumor. Treatment with VIP236 provided durable tumor regressions in NSCLC and CRC PDX models as well as in orthotopic metastatic TNBC PDX models. Importantly, a significant reduction of micrometastasis was observed in lung and brain tissue, providing a platform of evidence for the utility of VIP236 in metastatic disease. VIP236 is currently being evaluated in a first-in-human study in patients with advanced or metastatic solid tumors (NCT05712889).

## Data Availability

The data can be shared upon request.

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
