# Peer review of "Discovery of VIP236, an αvβ3-Targeted Small-Molecule–Drug Conjugate with Neutrophil Elastase-Mediated Activation of 7-Ethyl Camptothecin Payload for Treatment of Solid Tumors"

_cancers, 2023, doi:10.3390/cancers15174381_

Round 1
Reviewer 1 Report
Thank you for the opportunity to review the manuscript entitled "Discovery of VIP236, an αvβ3-Targeted Small Molecule-Drug Conjugate with Neutrophil Elastase-Mediated Activation of 7-Ethyl Camptothecin Payload for Treatment of Solid Tumors". I commend the authors for their comprehensive and well-written paper. However, I suggest some considerations and additional discussion areas to improve the manuscript further.
First, the role of neutrophil elastase in cleaving the linker and releasing the payload has been linked to a major side effect of ADCs, namely neutropenia. The manuscript reminded me that this neutrophil elastase-specific conjugate may carry a similar risk. More detailed safety data would be valuable to understand the potential impact.
Second, small molecule drug conjugates (SMDCs) typically have poor pharmacokinetics compared to ADCs. How does VIP236's pharmacokinetic profile compare? The manuscript would benefit from the addition of such data.
Third, the Val-Cit linker is known to be cleavable by esterases such as Ces1C. How does the linker used in this study compare? Even if the peptide portion is not cleaved, there is a possibility that the ester bond connecting the payload is susceptible to hydrolysis. It would be helpful to have additional discussion or data regarding the stability and cleavage susceptibility of the linker used.
Incorporating these aspects into the discussion will enhance the paper and may provide more comprehensive insight into the therapeutic potential and limitations of VIP236.
I look forward to seeing the revisions and learning more about this exciting development in the field of drug conjugates for cancer therapy.
Author Response
First, the role of neutrophil elastase in cleaving the linker and releasing the payload has been linked to a major side effect of ADCs, namely neutropenia. The manuscript reminded me that this neutrophil elastase-specific conjugate may carry a similar risk. More detailed safety data would be valuable to understand the potential impact.
Thank you for raising this important aspect. We are aware of this issue of all ADCs and particularly of vcMMAE ADCs, where neutropenia has been attributed to extracellular cleavage by neutrophil elastase. With the design of the SMDC with a significantly shorter half-life we were looking for an efficient activation by an enzyme abundant in TME to release the payload. Furthermore, we selected the 7-ethyl-camptothecin payload with lower potency than MMAE to reduce this risk of neutropenia and furthermore for high tumor penetration to minimize re-distribution. Taken together, the shorter half-life and the lower potency of the payload may reduce this risk of neutropenia.
In bone marrow, elastase is in intracellular vesicles of neutrophils in enzymatically inactive state and is activated only after secretion into extracellular space upon stimulation/activation of the immune cells. We would not expect cellular uptake of the SMDC and subsequent intracellular release of the payload due to the poor membrane permeability of the SMDC (Caco-2 flux assay: Papp A-B ~ 1nm/s).
Second, small molecule drug conjugates (SMDCs) typically have poor pharmacokinetics compared to ADCs. How does VIP236's pharmacokinetic profile compare? The manuscript would benefit from the addition of such data.
VIP236 SMDC indeed has shorter half-life in comparison to full-size antibodies, but it still shows low clearance and related high exposure as discussed in section 3.4 and table S2. Furthermore, as the efficacy is given due to accumulation of released free payload in the tumor, this could bring an advantage for VIP236 in comparison to other ADC approaches regarding potential toxic side effects. Due to impressive stability of VIP236 in plasma and other healthy tissues as well as biliary excretion of unchanged VIP236 (non-cytotoxic) form, systemic plasma concentration to free payload is very moderate (see Table 3 of the manuscript). These aspects are discussed a bit more in the discussion section in line 554 now.
Third, the Val-Cit linker is known to be cleavable by esterases such as Ces1C. How does the linker used in this study compare? Even if the peptide portion is not cleaved, there is a possibility that the ester bond connecting the payload is susceptible to hydrolysis. It would be helpful to have additional discussion or data regarding the stability and cleavage susceptibility of the linker used.
This is also a very good comment. The optimization of the SMDC was a multi-factor optimization which included stability, water solubility and elastase-dependent potency in vitro and high in vivo activity. In this context the ester bond connecting the linker peptide to the 7-EC payload is a key position, as it is the cleavage site for neutrophil elastase and on the other hand is potentially sensitive to unspecific hydrolytic cleavage. We will elaborate a bit more in the paper on the impact of the sterically demanding valine residue on hydrolytic stability (>79d in aqueous solution) and at the same time enabling efficient cleavage by neutrophil elastase (line 545). The sodium salt of the side chain carboxy group of aspartic acid in the linker significantly contributed to high solubility. Based on these aspects, employing the vc linker as a substrate for Ces1C would probably not meet our stability and solubility criteria of the SMDC.
Reviewer 2 Report
The presented manuscript covers a thorough investigation of the therapeutic effect of VIP236 as an αvβ3-targeted small molecule–drug conjugate. In this manuscript, the authors conducted in vitro and in vivo studies to prove the high efficacy of VIP236 to different tumor models via neutrophil elastase-mediated small drug release. Overall, this is a comprehensive research article with sufficient data that would attract wide interest in this field. Here attached my comments.
Major points:
1. The authors need to replace all the figures/tables/captions with a higher-resolution version.
2. The major concern of this design is that as the small molecule drug is released by neutrophil elastase, could the authors evaluate the risk of neutropenia in humans? The VIP236 may kill the neutrophils or its stem cells in bone marrow to cause severe side effects in clinical studies.
Minor points
3. I would suggest the authors put in the actual curve of in vitro cytotoxicity study in addition to the IC50 value table.
4. In Figure 6, could the authors also provide a bodyweight curve for the LXF529 model?
5. Do the authors have animal data for the small molecule drug only in any of the animal models used? Or could the authors provide literature data for it?
6. Please briefly explain the IHC score used in section 3.9.
7. Please use the consistent term “payload 1” across the manuscript.
8. Please pay attention to the font/size.
Check with font/size
Author Response
Major points:
- The authors need to replace all the figures/tables/captions with a higher-resolution version.
Thank you. We have submitted higher-resolution figures, tables, and captions.
- The major concern of this design is that as the small molecule drug is released by neutrophil elastase, could the authors evaluate the risk of neutropenia in humans? The VIP236 may kill the neutrophils or its stem cells in bone marrow to cause severe side effects in clinical studies.
Thank you for this important comment. The preclinical studies suggest an efficient payload release by NE from the SMDC in the tumor due to the 11-fold higher levels of 7-ethyl-camptothecin measured in the tumor versus plasma. NE is stored in intracellular vesicles, e. g. in neutrophils and is not enzymatically active in this state and thus could cleave the SMDC only if secreted to the extracellular space after stimulation/activation of the corresponding immune cells.
We may expect toxicity to neutrophils only after payload release mediated by elastase, which is excreted from neutrophils and macrophages in the tumor microenvironment. In bone marrow, however, elastase is in intracellular vesicles of neutrophils and not excreted extracellularly. We would not expect cellular uptake of the SMDC e. g. in bone marrow and subsequent intracellular release of the payload due to the poor membrane permeability of the SMDC (Caco-2 flux assay: Papp A-B ~ 1nm/s).
Minor points
- I would suggest the authors put in the actual curve of in vitro cytotoxicity study in addition to the IC50 value table.
Added as requested in supplementary information Figure S5.
- In Figure 6, could the authors also provide a bodyweight curve for the LXF529 model?
Added as requested in Figure 6B.
- Do the authors have animal data for the small molecule drug only in any of the animal models used? Or could the authors provide literature data for it?
- Application of VIP126 as SMDC VIP236 significantly improved response rate (66.7% vs 25%) compared to the equimolar dose of VIP126 given directly in the NCI H69 mouse model;
- Body weight loss as first sign of adverse effects is also significantly reduced (about 5-fold lower) when comparing VIP236 versus VIP126.
- Both improvements are based on VIP236 targeted delivery to the tumor.
|
Dose/Schedule |
Max Body Weight Loss (%) |
CR |
PR |
SD |
PD |
T/C volume |
Response Rate (%) |
|
Vehicle PBS |
-1 |
0 |
0 |
0 |
10 |
1.00 |
0.0 |
|
VIP236 20 mg/kg 2on/5off |
-1.21 |
0 |
8 |
2 |
0 |
0.04 |
66.7 |
|
Payload1 5 mg/kg 2on/5off |
-5.75 |
0 |
3 |
3 |
4 |
0.09 |
25.0 |
- Please briefly explain the IHC score used in section 3.9.
The score was given based on the number of positive cells. The methods have been updated to reflect this.
- Please use the consistent term “payload 1” across the manuscript.
This has been edited.
- Please pay attention to the font/size.
This has been edited.
Reviewer 3 Report
The authors compiled a comprehensive manuscript that details a novel SMDS candidate, VIP236 for targeting cancer therapy. The manuscript describes the synthesis, in vitro study, in vivo PK/PD and tumor study, that together prove the efficacy potential of the new drug candidate. This novel SMDC VIP236 was developed and optimized to bind to αVβ3 integrin on cancer cells and activated endothelial cells in aggressive or metastatic tumors and to release the modified CPT payload VIP126, which has improved solubility and cellular permeability, and is selectively cleaved by neutrophil elastase, which is highly expressed in the TME. Overall, the study is well-researched, supported by compelling evidences and provides valuable contribution to the field, so I would recommend acceptance of the manuscript with minor revisions.
1. Since this manuscript has focused heavily in PK study, the HPLC/LCMS bioanalysis assays in different buffers/biological matrices could be elaborated and validated (e.g., LC conditions, chromatograms showing specificity, sensitivity, justification of using tolbutamide as internal standard, etc.) Those could be added to the supplemental information.
2. Line 107, why VIP236 was dissolved in acetonitrile/DMSO for the plasma stability study as it is water soluble?
3. Hepatocytes stability (Line 304) study shows some loss in the VIP236 over 4 h incubation. Is it cleavage of the payload or other modes of degradation?
4. Line 133, for the in vitro cytotoxicity study, what cell lines are included in the panel?
5. Did you check the potential off target accumulation of VIP236 in other organs? What is the therapeutically relevant/safe concentration of the payload VIP126?
6. ANOVA followed by post-hoc analysis instead of t-test should be used for multiple group analysis.
Author Response
- Since this manuscript has focused heavily in PK study, the HPLC/LCMS bioanalysis assays in different buffers/biological matrices could be elaborated and validated (e.g., LC conditions, chromatograms showing specificity, sensitivity, justification of using tolbutamide as internal standard, etc.) Those could be added to the supplemental information.
Thank you for this comment. We believe that the details of the bioanalysis are not within the scope of this manuscript where we focus on discovery, synthesis, and mechanism of action of VIP236. The bioanalysis can be included in a future publication when we have clinical pharmacokinetics and drug metabolism from the human clinical trials.
- Line 107, why VIP236 was dissolved in acetonitrile/DMSO for the plasma stability study as it is water soluble?
This was the standardized protocol and matrix used in this assay to be compatible with a broad variety of test compounds.
- Hepatocytes stability (Line 304) study shows some loss in the VIP236 over 4 h incubation. Is it cleavage of the payload or other modes of degradation?
In the specific hepatocyte study mentioned, only degradation of parent (VIP236) was monitored. However, in other studies where higher concentrations of VIP236 were incubated, some preliminary metabolite identification was performed, and we do see the presence of the payload suggesting that cleavage does contribute to the limited loss of the parent molecule.
- Line 133, for the in vitro cytotoxicity study, what cell lines are included in the panel?
NCI-H292 and LoVo (see results section 3.3). Cell lines have been added to the methods and results.
- Did you check the potential off target accumulation of VIP236 in other organs? What is the therapeutically relevant/safe concentration of the payload VIP126?
Aside from tumor data, no detailed biodistribution data has been collected for VIP236. The therapeutic relevant exposures of VIP126 payload when delivered via VIP236 administration ranges from a VIP126 AUC of 179 to 920 ng*hr/mL for 2 days on a weekly schedule (2 on/5 off). This corresponds to average VIP126 plasma concentrations of 7.4 to 38 ng/mL over the 2 days that VIP236 was delivered. This average VIP126 plasma concentration range corresponds to a dose of 20 mg/kg 2 on/5 off shown in the Table below where there is robust tumor growth inhibition. These VIP126 concentrations are tolerated as the body weight loss in the specific dose group is <6%.
|
Dose/Schedule |
Max Body Weight Loss (%) |
T/C volume |
|
Vehicle PBS |
-1 |
1.00 |
|
VIP236 20 mg/kg 2on/5off |
-1.21 |
0.04 |
|
Payload1 5 mg/kg 2on/5off |
-5.75 |
0.09 |
- ANOVA followed by post-hoc analysis instead of t-test should be used for multiple group analysis.
Thank you for asking about the statistics we used. We performed the unpaired t-test at the end of the studies because the vehicle group was losing animals due to tumor size and group sizes were changing over time. For other comparisons we use Kruskal-Wallis Test (Dunn's multiple comparisons test) at the end of the treatment phase.
Round 2
Reviewer 1 Report
I want to suggest to accept this manuscript for publication.
Author Response
We thank the reviewer for their consideration of our manuscript.